# Metformin Ameliorates Testicular Function and Spermatogenesis in Male Mice with High-Fat and High-Cholesterol Diet-Induced Obesity

**DOI:** 10.3390/nu12071932

**Published:** 2020-06-29

**Authors:** Chin-Yu Liu, Ting-Chia Chang, Shyh-Hsiang Lin, Sheng-Tang Wu, Tai-Lung Cha, Chih-Wei Tsao

**Affiliations:** 1Department of Nutritional Science, Fu Jen Catholic University, Taipei 242, Taiwan; nf351.lab@gmail.com (C.-Y.L.); ctc5628@gmail.com (T.-C.C.); 2School of Nutrition and Health Sciences, College of Nutrition, Taipei Medical University, Taipei 110, Taiwan; lin5611@tmu.edu.tw; 3Master of Program of Food Safety, College of Nutrition, Taipei Medical University, Taipei 110, Taiwan; 4Division of Urology, Department of Surgery, Tri-Service General Hospital, National Defense Medical Center, Taipei 114, Taiwan; doc20283@gmail.com (S.-T.W.); tlcha@ndmctsgh.edu.tw (T.-L.C.)

**Keywords:** high-fat and high-cholesterol diet, obesity, male infertility, metformin, spermatogenesis

## Abstract

The aim of this study was to investigate the effects of metformin supplementation on metabolic dysfunction, testicular antioxidant capacity, apoptosis, inflammation and spermatogenesis in male mice with high-fat and high-cholesterol diet-induced obesity. Forty male C57BL/6 mice were fed a normal diet (NC group, *n* = 10) or a high-fat and high-cholesterol diet (HFC group, *n* = 30) for 24 weeks, and mice randomly chosen from the HFC group were later treated with metformin for the final 8 weeks of HFC feeding (HFC + Met group, *n* = 15). Compared with the HFC group, the obese mice supplemented with metformin exhibited improved blood cholesterol, glucose and insulin resistance. The HFC group diminishes in the sperm motility and normal sperm morphology, while the poorer maturity of testicular spermatogenesis was improved by metformin treatment. The HFC group exhibited a higher estradiol level and a lower 17β-HSD protein expression. Further analyses showed that metformin treatment increased the activities of superoxide dismutase, catalase and glutathione peroxidase and reduced lipid peroxidation. Nevertheless, both the HFC and HFC + Met groups exhibited increased expressions of apoptosis and inflammation proteins in the testis. Metformin treatment ameliorated obesity-induced poor testicular spermatogenesis and semen quality through increasing the testosterone level and antioxidant capacity.

## 1. Introduction

Infertility is a growing public health issue worldwide. The estimated number of couples that suffered from infertility increased from 42 million to 48.5 million from 1990 to 2010 [1]. On average, male infertility accounts for 40–50% of cases [2,3], and according to Agarwal et al., around 30 million men around the globe are infertile [4]. It was also found that sperm count, a predictable parameter by which to determine male infertility, remarkedly decreased between 1973 and 2011 [5]. The causes of male infertility are multifactorial, and obesity has emerged as an important related factor. Studies suggested negative impacts of obesity on sex hormones and sperm function [6,7]. Additionally, the odds of having hypogonadism, defined as a testosterone level lower than 300 ng/dL, was reported to be 2.74 times higher in the obese population [8]. Animal experiments have also highlighted reproductive dysfunction in high-fat diet-induced obesity, such as a disrupted testicular structure [9], sperm DNA damage [10] and poor spermatogenesis [11].

The potential mechanisms related to male infertility include oxidative stress, apoptosis and inflammation, and obesity can cause these adverse effects. A 3-year multicenter research trial revealed that obese, infertile men had significantly higher sperm DNA damage, which may be induced by oxidative stress [12]. Even with normal fertility, obese men were found to have an elevated reactive oxygen species (ROS) level and greater DNA fragmentation in sperm [13]. Other than oxidative stress, apoptosis plays important roles in sperm DNA damage and male infertility [14,15,16]. Inflammation is another factor that has a negative impact on sperm production, generates ROS and induces apoptosis [17].

The complete process of spermatogenesis is the key component in normal sperm quality and male fecundity. Briefly, spermatogonia located at the basement membrane of the seminiferous tubules enter mitosis and generate primary spermatocytes, which then undergo two meiotic divisions to form secondary spermatocytes and spermatids. The spermatids differentiate into mature spermatozoa through spermiogenesis [18]. The regulation and maintenance of spermatogenesis requires testosterone. Testosterone is the major androgenic hormone in male reproduction and is responsible for the development of secondary sexual characteristics, sexual desire and erectile function [19]. Besides, a normal testosterone level is essential to maintain bone density, muscle growth, the brain nervous system and cognitive health [20]. Testosterone production is regulated by the hypothalamic–pituitary–gonadal axis with appropriate levels of gonadotropin-releasing hormone (GnRH), follicle-stimulating hormone (FSH) and luteinizing hormone (LH) secretion. In testosterone biosynthesis, cholesterol is transported into Leydig cells by the steroidogenic acute regulatory protein (StAR). Cholesterol then transforms into pregnenolone, progesterone, androstenedione and testosterone through cytochrome P450 families and hydroxysteroid dehydrogenases (HSD) [21]. Despite cholesterol being the starting precursor for biosynthesis of steroid hormones including testosterone, studies have shown that a high plasma cholesterol level leads to a notable decrease in semen quality [22,23].

Metformin is a widely used drug for the treatment of type II diabetes and acts via lowering hepatic glucose production with insulin resistance and increasing the glucose uptake in muscle. Recent data have demonstrated the beneficial effects of metformin on weight loss, cancer, cardiovascular disease and polycystic ovary syndrome [24]. In terms of male fertility, metformin treatment administered to obese Wistar rats resulted in an improvement of semen quality [25]. A human study conducted in infertile patients with metabolic syndrome reported similar findings, with metformin increasing the androgen concentrations and sperm function [26].

Accordingly, the present study was performed to evaluate the effects of metformin against obesity-induced metabolic and reproductive damage. We hypothesized that metformin would improve the obesity-induced spermatogenesis deficiency and metabolic profile by suppressing potential related mechanisms including oxidative stress, apoptosis and inflammation.

## 2. Materials and Methods

### 2.1. Animals

All experimental procedures were approved by the Institutional Animal Care and Use Committee (IACUC; ethical code number: LAC-2016-0358) of Taipei Medical University (Taipei City, Taiwan). Male C57BL/6 mice (8-weeks-old, weighing 22–24 g) were procured from the National Laboratory Animal Center (Taipei City, Taiwan). The animals were housed under standard conditions (room temperature: 23 ± 2 °C; humidity: 55 ± 5%; 12-h light–dark cycles) and were given free access to food and water.

### 2.2. Experimental Design

After 2 weeks of acclimatization, the male C57BL/6 mice were randomly divided into two groups: the NC group (*n* = 10), which was given a normal chow diet (AIN-93G), and the HFC group (*n* = 30), which was administered a high-fat diet plus 1.5% (*w*/*w*) cholesterol. After a period of 16 weeks in which obesity was induced, the mice in the HFC group were randomly divided into two groups: one was maintained on the original diet (HFC, *n* = 15), and the other received a diet supplemented with 0.05% (*w*/*w*) metformin (HFC + Met, *n* = 15) for 8 weeks. The ingredients of each diet are shown in Table 1. Body weight and food intake from the beginning to the end of this experiment were documented. Blood samples were collected under anesthesia for further biochemical analyses. Liver and testis sections were partly fixed in 10% formalin (diluted from 37% formaldehyde solution, J.T. Baker, Phillipsburg, NJ, USA) for morphological analysis, and the rest were frozen in liquid nitrogen. Semen samples were obtained immediately after the vas deferens was removed for assessment of mouse sperm parameters including sperm motility, sperm count and morphological abnormalities.

### 2.3. Histological Analysis

Formalin-fixed liver and testicular tissue samples were treated at the Department of Pathology of Cardinal Tien Hospital (New Taipei City, Taiwan), cut into sections, and stained with Hematoxylin and Eosin (H&E). Then, the liver and testicular tissue were observed and photographed under 40×, 100× and 400× magnification using a system incorporated into an ergonomic system microscope (DM1000, Leica, Wetzlar, Germany). The thickness of the germinal epithelium and the mean seminiferous tubule diameter (MSTD) were calculated using Image J software (1.50, National Institutes of Health, Bethesda, MD, USA). Testicular spermatogenesis was determined according to Johnsen’s score [27].

### 2.4. Serum Analysis

Serum was centrifuged for 20–30 min at 2000× *g* and isolated. Serum glucose (GLU), total cholesterol (TC), alanine aminotransferase (ALT) and aspartate aminotransferase (AST) level analyses were performed using a hematologic instrument (ProCyte Dx, IDEXX, Westbrook, MA, USA). The serum triglycerides (TG) level was measured using a biochemical analyzer (DRI-CHEM 3500s, Fuji, Tokyo, Japan). The serum insulin level was determined via an Enzyme-Linked Immunosorbent Assay (ELISA) using a commercial ELISA kit according to the manufacturer’s instructions. The homeostasis model assessment of insulin resistance (HOMA-IR) [28] was estimated using the following formula:HOMA-IR = fasting blood glucose (nmol/L) × fasting serum insulin (μU/mL)/22.5

### 2.5. Semen Quality Analysis

Sperm motility, which is represented as the percentage of motile sperm, was evaluated microscopically under a magnification of 40×. The numbers of motile and nonmotile sperm were counted in 4 random microscopic fields, and at least 200 sperm cells were counted. The sperm count was examined using an automated cell counter (TC20, Bio-Rad, Hercules, CA, USA). In the assessment of sperm morphology, slides of sperm samples were dry-prepared, fixed with methanol (Honeywell, Morris Plains, NJ, USA), and stained with a mixture of Eosin Y (E4009, Sigma-Aldrich, Saint Louis, MO, USA) and ethanol (Bioman, Taipei City, Taiwan). Then, slides were rinsed with 75% ethanol (Bioman, Taipei City, Taiwan) and dried, and the percentage of normal sperm in a minimum of 100 spermatozoa was calculated.

### 2.6. Testicular Cholesterol and Sex Hormone Analysis

Frozen testis tissues were thawed and homogenized in ice-cold radio immunoprecipitation assay (RIPA) lysis buffer (Thermo Fisher Scientific, Waltham, MA, USA) containing protease inhibitors and phosphatase inhibitors and centrifuged at 14,000× *g* (4 °C) for 20 min to obtain the supernatant. The testicular concentrations of cholesterol, estradiol and testosterone were measured using commercial ELISA kits according to the manufacturer’s instructions (MyBioSource, MBS164208, San Diego, CA, USA; MyBioSource, MBS261250, San Diego, CA, USA; Cayman, Item No. 582701, Ann Arbor, MI, USA).

### 2.7. Western Blotting Analysis

Testis samples were prepared as described previously. Protein concentrations were quantified using a detergent compatible protein assay (Bio-Rad). Sufficient protein samples were resolved using sodium dodecyl sulfate-polyacrylamide gels (Bioman, Taipei City, Taiwan) and transferred to polyvinylidene difluoride (PVDF) membranes (GE Healthcare, Freiburg, Germany). Then, the membranes were blocked with 5% dried milk in Tris-buffered saline with 0.1% Tween 20(TBST) for 1 h and incubated overnight with primary antibodies. After incubation, the membranes were washed with TBST, incubated with anti-mouse IgG (1:5000; sc-2005, Santa Cruz Biotechnology, Dallas, TX, USA) or anti-rabbit IgG (1:4000; sc-2054, Santa Cruz Biotechnology) for 1 h at room temperature, and visualized using an ECL kit (Omicsbio, Taipei City, Taiwan). Then, the following primary antibodies that detect enzymes involved in testosterone biosynthesis, apoptosis and inflammation were used for western blotting: StAR (1:1000; sc-25806, Santa Cruz Biotechnology), 3β-HSD (1:500; sc-28206, Santa Cruz Biotechnology), 17β-HSD (1:250; sc-135044, Santa Cruz Biotechnology), CYP11A1 (1:1000; sc-202456, Santa Cruz Biotechnology) and CYP17A1 (1:1000; sc-66850, Santa Cruz Biotechnology), Caspase 8 (1:1000; 59607, GeneTex, San Antonio, TX, USA), Cytochrome C (1:1000; sc-13156, Santa Cruz Biotechnology), Bax (1:1000; 2772, Cell Signaling Technology, Danvers, MA, USA), Caspase 9 (1:1000; 9508, Cell Signaling Technology), Caspase 3 (1:500; 9662, Cell Signaling Technology), Cleaved-Caspase 3 (1:250; 9664, Cell Signaling Technology), poly (ADP-ribose) polymerase (PARP;1:1000; 3542, Cell Signaling Technology), Bcl-xl (1:1000; ab32370, Abcam, Cambridge, MA, USA), peroxisome proliferator-activated receptor ϒ (PPARϒ; 1:1000; sc-7273, Santa Cruz Biotechnology), interleukin-6 (IL-6; 1:1000; sc-57315, Santa Cruz Biotechnology), tumor necrosis factor-α (TNF-α; 1:1000; ab1793, Abcam) and nuclear factor-κB (NF-κB; 1:1000; E381, Abcam). The intensity of each band was quantified using Image J software and normalized to the expression of β-actin (1:10,000; A5316, Sigma-Aldrich).

### 2.8. Statistical Analysis

All obtained data were expressed as means ± S.D. Mean values among the three groups were compared using one-way analysis of variance (ANOVA) followed by Fisher’s protected least significant difference test (LSD). Differences were defined as statistically significant if the *p*-value was less than 0.05. The statistical analyses were conducted using SAS software, version 9.4 (SAS Institute Inc., Cary, NC, USA).

## 3. Results

### 3.1. Metformin Ameliorates the Glucose Level and HOMA-IR But Not Body Weight and Lipid Profiles

There were no statistical differences of daily food intake among the individual groups (NC: 4.1 ± 0.6 g, HFC: 3.3 ± 0.6 g and HFC + Met: 2.6 ± 0.3 g). To evaluate the metabolic effect of metformin, 0.05% metformin was administered for 8 weeks and the actual intake was about 30 mg/kg/day calculated by actual intake. The mice fed a high-fat and high-cholesterol diet (HFC) had significantly higher body weight, serum glucose, HOMA-IR and total cholesterol levels. In the high-fat diet supplemented with metformin group (HFC + Met), blood sugar, insulin resistance and the total cholesterol level were improved whereas no change was found in body weight. However, following metformin treatment, there was an increase in serum cholesterol as compared with the mice fed a normal diet (NC). The serum insulin and TG levels were similar among the three groups (Figure 1).

### 3.2. Metformin Treatment Reverses neither Liver Weight nor Liver Steatosis in Obese Mice

Compared with the NC group, the mice in the HFC group exhibited marked increases in the absolute and relative liver weights. As presented in the figures showing liver-section histology, HFC feeding resulted in hepatic steatosis, with increased fat vacuoles in hepatocytes and an enlarged lipid droplet size. In addition, the serum concentrations of ALT and AST, markers of liver injury, in the HFC-fed mice were notably increased by approximately 2.7-fold and 1.8-fold, respectively, relative to the control group. Metformin treatment administered to the HFC-fed obese mice reversed neither the significant increases in liver weight and the liver weight to body weight ratio nor liver steatosis, as indicated by the likenesses of the liver-section histology results. Consistently, the ALT and AST levels were much greater in the metformin-treated mice than in the control group (Figure 2).

### 3.3. Metformin Treatment Improves the Testicular Spermatogenesis and Semen Quality in Obese Mice

The reproductive organs, including the testis, epididymis and vas deferens, were collected and weighed. Unlike the body weight and liver weight, no differences were found in the absolute weight of the reproductive organs between the NC and HFC groups; however, the means of the relative organ weights were reduced in the HFC group. On the other hand, the testis and epididymis absolute weights as well as the ratios of reproductive organ to body weight of the metformin-treated mice did not differ from those of the HFC-fed mice, while the mean vas deferens absolute weight in the HFC + Met group was lower than the weights of the other groups. At the same time, compared with the NC group, the semen quality of the HFC group was significantly decreased; however, with the exception of sperm count, the mice treated with metformin exhibited improved sperm motility and a greater percentage of normal morphology (Figure 3).

In the present study, abnormal testicular histology and semen quality were found to be important characteristics of sterility. Testicular morphology analysis of the HFC-fed mice revealed incomplete spermatogenesis, with disorderly arranged germ cells, a reduced number of mature sperm and a thinner germinal epithelium, whereas following metformin treatment, as shown in the figure, the testes of the mice had organized spermatogenic cells with increased mature spermatozoa. The morphometric analyses also indicated that the HFC group had a lower mean testicular biopsy score (MTBS) as compared with the NC group and the HFC + Met group (Figure 4).

### 3.4. Metformin Acts to Increase Testosterone Level in Obese Mice through Upregulating 17β-HSD Expression

The testicular cholesterol, testosterone and estradiol levels were measured by ELISA. Comparing the mice fed a normal chow diet and those fed a high-fat diet, higher testicular cholesterol and estradiol levels along with a lower testosterone level were observed in the HFC-fed mice. In contrast, the mice treated with metformin exhibited a remarkable increase in the testosterone concentration and a decrease in the estradiol level (Figure 5).

A similar decreasing trend was also observed in the testicular cholesterol level, but this did not reach statistical significance (Figure 5). Subsequently, the testosterone biosynthesis pathway and the protein expressions of related molecules were determined. Compared with the protein expression in the HFC group, 17β-hydroxysteroid dehydrogenase (17β-HSD), which catalyzes the conversion of androstenedione into testosterone, was significantly upregulated in the HFC + Met group and did not differ from the level in the NC group (Figure 6).

### 3.5. Metformin Suppresses the Oxidative Stress Level in the Testis but Dose Not Reverse the Testicular Apoptosis and Inflammation

The present study examined the activities of antioxidant enzymes and the production of malondialdehyde (MDA) as markers of antioxidant status and oxidative damage in the testis. The results showed that the SOD, CAT and GPx activities in the metformin-treated mice were clearly higher than in the HFC-fed obese mice. Additionally, the level of lipid peroxidation product MDA was significantly different among the three groups: the highest level was observed in the HFC-fed obese mice, the second highest was in the metformin-treated obese mice and the lowest was in the control group (Figure 7). These results indicated that metformin alleviated high-fat and high-cholesterol-diet-induced excess oxidative stress in the testes.

To further verify the potential mechanism of high-fat diet-induced testicular and sperm dysfunction, the protein levels of apoptosis- and inflammation-related molecules were detected using western blotting. In the high-fat and high-cholesterol diet-fed mice, the protein expressions of apoptosis-related mediators (ratio of Bax and Bcl-xl, cleaved form of caspase 3 and PARP) were significantly increased (Figure 8). Moreover, similar changes were observed in inflammation-associated mediators (TNF-α and NF-κB), and metformin supplementation did not reverse the testicular apoptosis and inflammation (Figure 9).

## 4. Discussion

Obesity has been thought to be associated with various diseases, including male infertility. Following administration of a 24-week high-fat and high-cholesterol diet, male C57BL/6 mice developed obesity, hyperglycemia, insulin resistance and hypercholesterolemia. Similar to findings obtained in other studies [29,30,31], metformin treatment ameliorated hyperglycemia and insulin resistance; the minor improvement of hypercholesterolemia was also observed between the groups of HFC and HFC + Met. However, the mean final body weight of the obese mice did not differ significantly, whether treated with metformin or not. Seifarth et al. [32] reported that the effectiveness of weight loss in the obese is dependent on the severity of insulin resistance. Also, a systematic review revealed that obese patients with a BMI greater than 35 kg/m^2^ may have greater weight loss with metformin use [33]. Besides, the differences in the mean serum TG level among the three groups were not significant. The possible mechanism contributing to this result may be related to the upregulation of cholesterol absorption in the liver due to a high dietary cholesterol intake [34].

On a high-fat and high-cholesterol diet, the mice also developed a higher liver weight and higher serum levels of ALT and AST in addition to hepatic steatosis. Nevertheless, unlike in other rodent models [35,36], effects of metformin in terms of reducing liver weight and improving serum parameters of liver function as well as histological changes were not observed. Human studies and meta-analyses have presented similar findings in terms of failure of metformin treatment to improve obesity-related hepatic damage [37,38,39]. The practice guidelines provided by the American Association for the Study of Liver Diseases also excluded the use of metformin in nonalcoholic fatty liver disease (NAFLD) [40]. Meanwhile, long-term (6 months) metformin treatment in a NAFLD rat model did not reverse the development of fatty liver. The probable cause was that genes associated with the metabolism of glucose and cholesterol were regulated, but fatty acids were not [41], which in turn is consistent with the results that metformin use did not alter the liver damage but did improve the high blood sugar and cholesterol levels.

As metformin has potential as a therapeutic application in obesity, the effect of metformin on HFC-induced male infertility was investigated. In the present study, a high-fat and high-cholesterol diet resulted in lower relative reproductive organ weights and imbalances in reproductive hormones. Moreover, testicular morphological injury and reduced sperm function were also noted in the obese mice. Consistent with previous reports [42,43,44], upon metformin treatment, the hormone concentrations, integrity of sperm production, maturity of sperm and semen quality were reversed; however, the change in reproductive organ weight was not, which was more likely affected by the body weight gain. Appropriate hormone levels are crucial for male reproductive capacity. Testicular testosterone concentration plays an important part in spermatogenesis [45], and thus, the levels of testosterone and cholesterol, the precursor of testosterone, were measured in this study. Similar to the biochemical value, the intratesticular cholesterol concentration was increased in the obese group and decreased after metformin treatment but did not reverted to normal. Morgan et al. reported that a high-cholesterol diet in rabbits resulted in a higher plasma cholesterol concentration and more lipid deposition in the seminiferous tubules, which may be attributed to the excessive dietary cholesterol intake and hence increased cholesterol transport into the testis [46]. However, mice with a high intratesticular cholesterol level had a lower testosterone concentration and a lower protein expression of 17β-hydroxysteroid dehydrogenase, an enzyme involved in the testosterone formation pathway that converts androstenedione to testosterone. Similarly, previous studies showed that a higher dietary cholesterol intake led to histological abnormalities in testicular Leydig cells and may disrupt the production of testosterone [47]. As androstenedione is the precursor of both testosterone and estradiol and obesity is known to be associated with hormonal changes [48,49], the estradiol concentration was measured. The results suggested that long-term excessive fat and cholesterol intakes decreased the testosterone production and increased the estradiol level, which were altered after metformin supplementation. Moreover, metformin treatment increased the protein expression of 17β-hydroxysteroid dehydrogenase. The possible processes that improve the male fertile capacity driven by metformin seem to be multifactorial. Rice et al. [50] and Fuhrmeister et al. [51] found that metformin inhibited the aromatase expression and decreased the formation of estradiol. On the other hand, it was reported that metformin may protect spermatogenesis and sperm function through activation of AMPK(AMP-activated Protein Kinase) signaling [52,53]. AMPK can also increase lactate production, which is involved in the ATP synthesis of germ cells and can downregulate germ cell death in spermatogenesis [54].

Given that male infertility is mainly associated with oxidative stress, apoptosis and inflammation in the testis and sperm [16], the effects of metformin on the pathway mentioned above were investigated. The results showed that metformin ameliorated the imbalance in the antioxidant system and decreased the testicular lipid peroxidation (MDA) level in the HFC-fed obese mice. An in vivo experiment conducted in the rats with high-fat food-induced obesity revealed that metformin intervention improved the contents of SOD and GSH-Px and decreased the content of MDA in the testis [55]. The use of metformin may activate the AMPK pathway, increasing alanine production and thus promoting the antioxidant capacity in the testis [54]. Metformin also improved HFD-induced obesity and lipid metabolism by the downstream molecules of AMPK [56]. In addition, in contrast with the results of other studies [57,58,59], metformin treatment did not exert antiapoptotic or anti-inflammatory action in this obese rodent model due to the elevations of apoptosis and inflammation mediators. Examination of a model of autoimmune orchitis (chronic inflammation and infertility) suggested that increasing the content of TNF-α may induce apoptosis of sperm [60].

## 5. Conclusions

In conclusion, our study demonstrated that metformin protected the testicular histology and semen quality disrupted in the obese mice model via increasing the testosterone concentration and enhancement of antioxidative enzymes in the testes.

## Figures and Tables

**Figure 1 nutrients-12-01932-f001:**
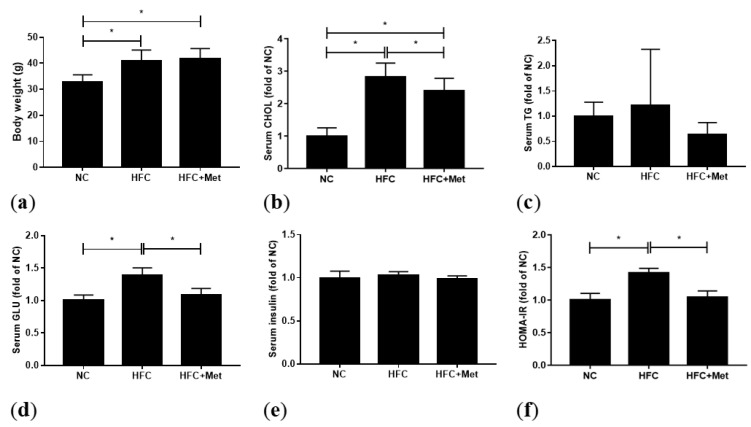
Effects of the high-fat and high-cholesterol diet and metformin supplementation on (**a**) body weight, (**b**) serum total cholesterol (CHOL), (**c**) serum triglycerides (TG), (**d**) serum glucose (GLU), (**e**) serum insulin level and (**f**) homeostasis model assessment of insulin resistance (HOMA-IR) in the male mice: Data are expressed as means ± SD. ((**a**): *n* = 10–15; (**b**–**f**): *n* = 4–8 per group). * *p* < 0.05 as determined by one-way ANOVA followed by least significant difference test (LSD) post hoc test. NC, normal diet; HFC, high-fat and high-cholesterol diet; HFC + Met, HFC with metformin supplementation.

**Figure 2 nutrients-12-01932-f002:**
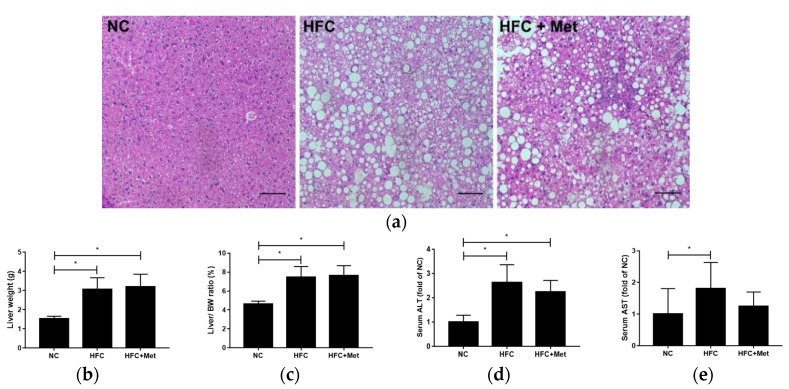
Effects of the high-fat and high-cholesterol diet and metformin supplementation on (**a**) hepatic histology, (**b**) liver weight, (**c**) liver to body weight ratio, (**d**) serum alanine aminotransferase (ALT) and (**e**) serum aspartate aminotransferase (AST) in the male mice: Hepatic sections were stained with hematoxylin and eosin. Data are expressed as means ± SD ((**a**): *n* = 6; (**b**–**e**): *n* = 9–15 per group). * *p* < 0.05 as determined by one-way ANOVA followed by LSD post hoc test. NC, normal diet; HFC, high-fat and high-cholesterol diet; HFC + Met, HFC with metformin supplementation. Scale bar: 200 μm.

**Figure 3 nutrients-12-01932-f003:**
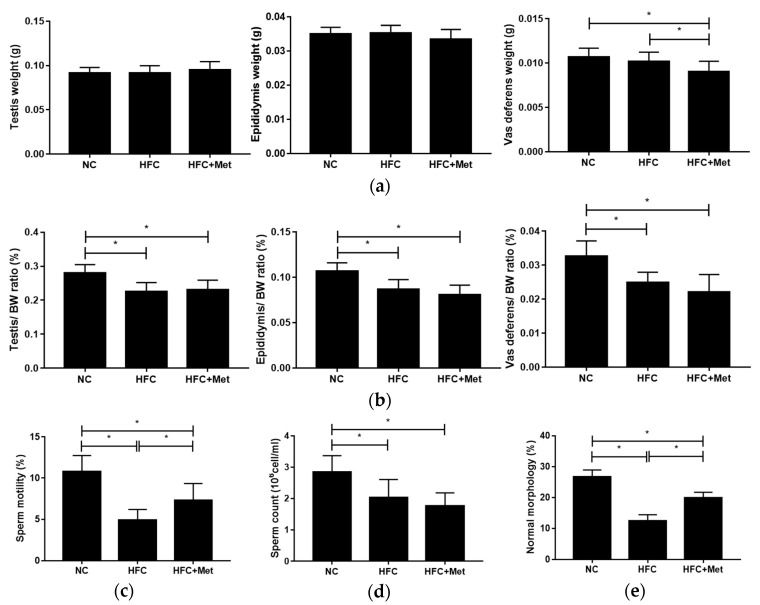
Effects of the high-fat and high-cholesterol diet and metformin supplementation on (**a**) the weights of the testis, epididymis and vas deferens; (**b**) the relative weights of the testis, epididymis and vas deferens; (**c**) sperm motility; (**d**) sperm count; and (**e**) percentage of normal sperm morphology in the male mice: Data are expressed as means ± SD (*n* = 9–15 per group). * *p* < 0.05 as determined by one-way ANOVA followed by LSD post hoc test. NC, normal diet; HFC, high-fat and high-cholesterol diet; HFC + Met, HFC with metformin supplementation.

**Figure 4 nutrients-12-01932-f004:**
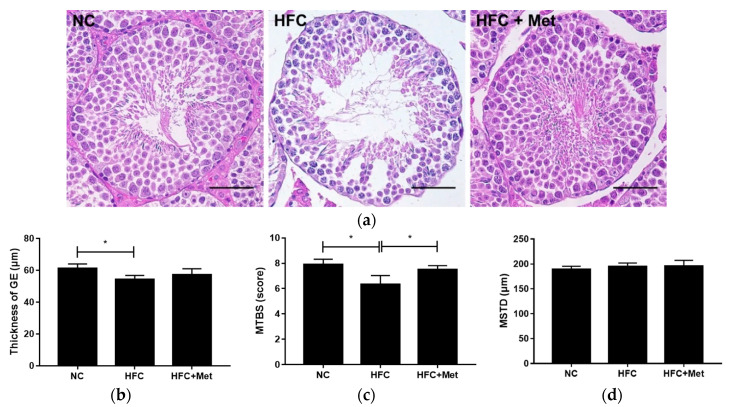
Effects of the high-fat and high-cholesterol diet and metformin supplementation on (**a**) testicular histology, (**b**) thickness of the germinal epithelium (GE), (**c**) mean testicular biopsy score (MTBS) and (**d**) mean seminiferous tubule diameter (MSTD) in the male mice: Testicular sections were stained with hematoxylin and eosin. Data are expressed as means ± SD (*n* = 5 per group). * *p* < 0.05 as determined by one-way ANOVA followed by LSD post hoc test. NC, normal diet; HFC, high-fat and high-cholesterol diet; HFC + Met, HFC with metformin supplementation. Scale bar: 50 μm.

**Figure 5 nutrients-12-01932-f005:**
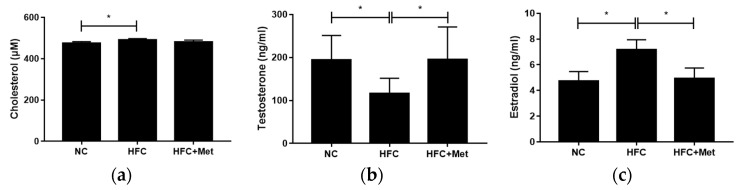
Effects of the high-fat and high-cholesterol diet and metformin supplementation on testicular (**a**) cholesterol, (**b**) testosterone and (**c**) estradiol levels in the male mice: Data are expressed as means ± SD (*n* = 5 per group). * *p* < 0.05 as determined by one-way ANOVA followed by LSD post hoc test. NC, normal diet; HFC, high-fat and high-cholesterol diet; HFC + Met, HFC with metformin supplementation.

**Figure 6 nutrients-12-01932-f006:**
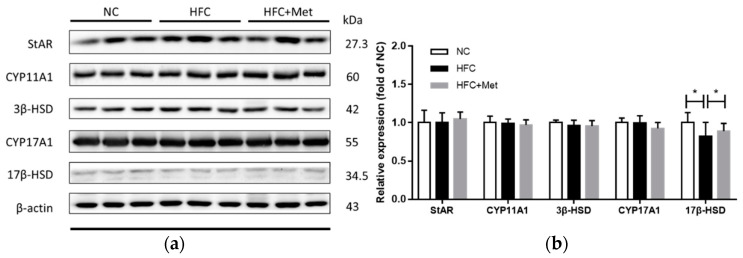
Effects of the high-fat and high-cholesterol diet and metformin supplementation on testosterone biosynthesis regulators (StAR, CYP11A1, 3β-HSD, CYP17A1 and 17 β-HSD) in the male mice: (**a**) Protein expressions in each group were measured using western blot analysis, and β-actin was used for normalization; (**b**) relative density analysis of the protein bands. Data are expressed as means ± SD of duplicate experiments (*n* = 6–9 per group). * *p* < 0.05 as determined by one-way ANOVA followed by LSD post hoc test. NC, normal diet; HFC, high-fat and high-cholesterol diet; HFC + Met, HFC with metformin supplementation.

**Figure 7 nutrients-12-01932-f007:**
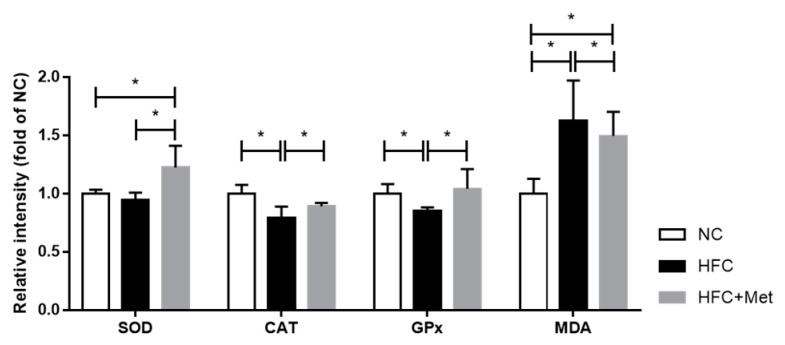
Effects of the high-fat and high-cholesterol diet and metformin supplementation on testicular antioxidants (superoxide dismutase (SOD), catalase (CAT) and glutathione peroxidase (GPx) activity) and malondialdehyde (MDA) content in the male mice: Data are expressed as means ± SD of duplicate experiments (*n* = 6–9 per group). * *p* < 0.05 as determined by one-way ANOVA followed by LSD post hoc test. NC, normal diet; HFC, high-fat and high-cholesterol diet; HFC + Met, HFC with metformin supplementation.

**Figure 8 nutrients-12-01932-f008:**
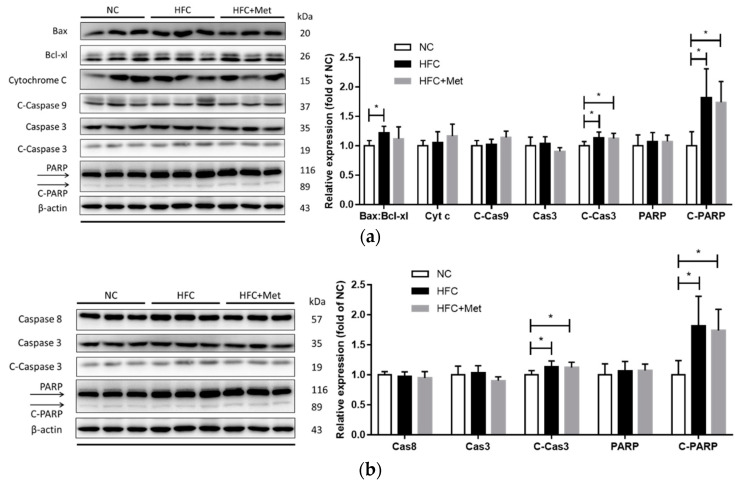
Effects of the high-fat and high-cholesterol diet and metformin supplementation on testicular intrinsic apoptosis pathway regulators, including (**a**) the ratio of Bax:Bcl-xl, cytochrome c (Cyt c), cleaved-caspase 9 (C-Cas9), caspase 3 (Cas3), cleaved-caspase 3 (C-Cas3), poly (ADP-ribose) polymerase (PARP) and cleaved-PARP (C-PARP) and (**b**) caspase 8 (Cas8), caspase 3 (Cas3), cleaved-caspase 3 (C-Cas3), PARP and cleaved-PARP (C-PARP) in the male mice: Protein expressions in each group were measured using western blot analysis, and β-actin was used for normalization. Data are expressed as means ± SD of duplicate experiments (*n* = 6–9 per group). * *p* < 0.05 as determined by one-way ANOVA followed by LSD post hoc test. NC, normal diet; HFC, high-fat and high-cholesterol diet; HFC + Met, HFC with metformin supplementation.

**Figure 9 nutrients-12-01932-f009:**
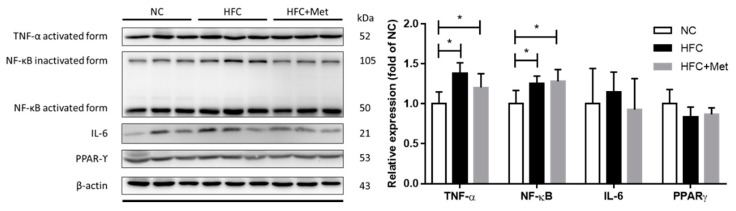
Effects of the high-fat and high-cholesterol diet and metformin supplementation on testicular inflammation pathway regulators (TNF-α, NF-κB, IL6 and PPARϒ) in the male mice: Protein expressions in each group were measured using western blot analysis, and β-actin was used for normalization. Data are expressed as means ± SD of duplicate experiments (*n* = 6–9 per group). * *p* < 0.05 as determined by one-way ANOVA followed by LSD post hoc test. NC, normal diet; HFC, high-fat and high-cholesterol diet; HFC + Met, HFC with metformin supplementation. TNF-α, tumor necrosis factor-α; NF-κB, nuclear factor-κB; IL-6, interleukin-6; peroxisome PPARϒ, proliferator-activated receptor ϒ.

**Table 1 nutrients-12-01932-t001:** Ingredients of the normal diet, the high-fat and high-cholesterol diet, and the high-fat and high-cholesterol diet with metformin supplementation.

**Ingredients**	**NC**	**HFC**	**HFC + Met**
Corn starch	41.0	29.5	29.5
Dextrin	15.5	10.0	10.0
Sucrose	10.0	10.0	10.0
Cellulose	5.0	5.0	5.0
Casein	19.0	19.0	19.0
Soybean oil	4.0	20.0	20.0
Mineral mix	3.5	3.5	3.5
Vitamin mix	1.0	1.0	1.0
Choline	0.25	0.25	0.25
Cholesterol	-	1.5	1.5
Cysteine	0.18	0.18	0.18
TBHQ	0.25	0.0008	0.0008
Metformin	-	-	0.05
**Energy (%)**	**NC**	**HFC**	**HFC + Met**
Carbohydrate	70	43	43
Fat	10	40	40
Protein	20	17	17

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
