# Peer review of "Metformin Ameliorates Testicular Function and Spermatogenesis in Male Mice with High-Fat and High-Cholesterol Diet-Induced Obesity"

_nutrients, 2020, doi:10.3390/nu12071932_

Round 1

Reviewer 1 Report

The paper Metformin ameliorates testicular function and spermatogenesis in male mice with high-fat and highcholesterol diet-induced obesity by Chin-Yu Liu et al., describe the results of treating obese mice with metformin in fertility
parameters (testicular function, spermatogenesis) and they also investigate some other biomarkers of oxidative stress,
inflammation and others. The conclusion of this work is that metformin ameliorates obesity-induced reduced testicular
spermatogenesis and semen quality by increasing testosterone levels and reducing oxidative stress.

I see no big difference with the previous version of the paper. The authors have clirify some bits, but overall it remains the same. I still cant understand the statistics. These letter should be substituted by asterisk, and bars should be pointing out what the saterisk refers to.

Author Response

Assistant Editor

Nutrients

Dear Professor Nemanja Marinkovic:

Please find enclosed our revised original paper entitled “Metformin ameliorates testicular function and spermatogenesis in male mice with high-fat and high-cholesterol diet-induced obesity”. We appreciated the comments and suggestions provided to further improve our manuscript.

Sincerely yours

Corresponding author: Chih-Wei Tsao, M.D., Ph.D.

Division of Urology, Department of Surgery,

Tri-Service General Hospital,

No. 325, Section 2, Cheng-Gung Road,

Neihu District, Taipei 11490, Taiwan, China.

Telephone: +886-2-87927170            Fax: +886-2-87927172

e-mail: weisurger@gmail.com

Comments of Reviewer 1 to Author:

The paper Metformin ameliorates testicular function and spermatogenesis in male mice with high-fat and highcholesterol diet-induced obesity by Chin-Yu Liu et al., describe the results of treating obese mice with metformin in fertility
parameters (testicular function, spermatogenesis) and they also investigate some other biomarkers of oxidative stress,
inflammation and others. The conclusion of this work is that metformin ameliorates obesity-induced reduced testicular
spermatogenesis and semen quality by increasing testosterone levels and reducing oxidative stress.

I see no big difference with the previous version of the paper. The authors have clirify some bits, but overall it remains the same. I still cant understand the statistics. These letter should be substituted by asterisk, and bars should be pointing out what the saterisk refers to.

Our response:

We thank the reviewer for the academic suggestion. We have revised all the legends of figures by replacing the presented letters with the asterisks. The changes of content were highlighted in red color.

Reviewer 2 Report

The authors have improved the manuscript and now it is clearer.

Author Response

Assistant Editor

Nutrients

Dear Professor Nemanja Marinkovic:

Please find enclosed our revised original paper entitled “Metformin ameliorates testicular function and spermatogenesis in male mice with high-fat and high-cholesterol diet-induced obesity”. We appreciated the comments and suggestions provided to further improve our manuscript.

Sincerely yours

Corresponding author: Chih-Wei Tsao, M.D., Ph.D.

Division of Urology, Department of Surgery,

Tri-Service General Hospital,

No. 325, Section 2, Cheng-Gung Road,

Neihu District, Taipei 11490, Taiwan, China.

Telephone: +886-2-87927170            Fax: +886-2-87927172

e-mail: weisurger@gmail.com

Comments of Reviewer 2 to Author:

The authors have improved the manuscript and now it is clearer.

Our response:

We thank the reviewer for the helpful comment to improve the quality of our manuscript.

This manuscript is a resubmission of an earlier submission. The following is a list of the peer review reports and author responses from that submission.

Round 1

Reviewer 1 Report

Brief summary

Chin-Yu Liu and co-workers study the metformin treatment on altered testicular function induced by high fat and cholesterol (HFC) diet. Metformin improved blood glucose, insulin resistance and the total cholesterol level. Metformin also improved sperm motility and a greater percentage of normal morphology. Testosterone concentration decreased and estradiol concentration increased in testes from mice fed a HFC diet.  Metformin treatment reverted these effects. The authors conclude metformin treatment increased antioxidant capacity in the testis, and this effect along with an increase in testosterone concentration protected the testicular histology and semen quality. However, metformin did not revert liver damage or testicular apoptosis and inflammation induced by HFC diet.

Broad comments

It is a well-designed paper with correct methodology. The manuscript is clear and quite complete; however I have some suggestions.

Blood total cholesterol level remained increased compare to control group after metformin treatment. It would be interesting to determine blood levels of very low-density lipoprotein (VLDL), low density lipoprotein (LDL) and high density lipoprotein (HDL) cholesterol. Even oxidation of LDL cholesterol (oxLDL) since it has more affinity for macrophages and induces inflammation. The article would improve with these measurements.

Increased testicular cholesterol in HFC group was not reverted with metformin treatment. There is a decreasing trend without statistical significance. Do the authors have any explanation? Can it be related to treatment concentration or duration? Have the authors performed experiments with a higher metformin concentration or treatment duration? Maybe, in that case, this trend would be significant.  

Metformin treatment increased antioxidant capacity in the testis, but MDA continued higher compared to control. Do the authors have any explanation? Can it be related to treatment concentration or duration?

Specific comments

Abstract: Line 27 “superoxide peroxidase”. It should be written superoxide dismutase.

Introduction: Lines from 63 to 71. It is general knowledge. This paragraph should be shortened. Moreover, refence 22 is not appropriate.

Could please the authors write a hypothesis at the end of the introduction?

Materials and Methods: Line 100 (HFC+Met, n=15), however in the abstract line 21 (HFC+Met group, n=16) Which one is the right number?

Results: Lines 241 and 242. At the end of the sentence: “A similar decreasing trend was also observed in the testicular cholesterol level, but this did not reach statistical significance” should indicate “table 2”.

Lines 264 and 265: The results showed that the SOD, CAT and GPx activities in the metformin-treated mice were higher than in the high-fat diet-fed mice. However, SOD activity is the same in control and HFC. Do the authors have any explanation?

Although metformin increased some antioxidant defences in the testis, it was not able to revert the increment of MDA levels induced by high fat diet as figure 6 shows. Can the authors explain why? How do the authors conclude the metformin has an antioxidant capacity in lines 31 and 370 if treatment with metformin did not decrease malondialdehyde content to control levels (figure 6)?

Discussion: In line 305, authors wrote: “metformin treatment ameliorated hypercholesterolemia, hyperglycaemia and insulin resistance”. However, serum cholesterol levels continue increased after metformin treatment compared to the control group (Figure 1b). Moreover, testicular cholesterol level is also increased in HFC and HFC + Met (table 2). Authors explain there is a decreasing trend without statistical significance (lines 241 and 242) and conclude in lines 335 and 336 that metformin decreases intratesticular cholesterol concentration. This conclusion is not appropriate since there are no statistical differences, they should improve it.

Lines 357-358: “The results showed that metformin ameliorated the imbalance in the antioxidant system and decreased the testicular lipid peroxidation (MDA) level in the HFC-fed obese mice” Is this of statistical significance? Figure 6 shows MDA levels increased in HFC+ Met compare to control.

Lines 361-362: “The use of metformin may activate the AMPK pathway, increasing alanine production and thus promoting the antioxidant capacity in the testis [55]”. The authors did not analyse this pathway, and it could be interesting since AMPK is a negative regulator of NADPH oxidase. Moreover, PPARg can increase AMPK. The authors measured PPARg protein expression and they did not find changes.  The article would benefit from measurement of AMPK or NAPDH oxidase protein level and elucidate the antioxidant capacity of metformin.

Author Response

Assistant Editor

Nutrients

Dear Professor Nemanja Marinkovic:

Please find enclosed our revised original paper entitled “Metformin ameliorates testicular function and spermatogenesis in male mice with high-fat and high-cholesterol diet-induced obesity”. We appreciated the comments and suggestions provided to further improve our manuscript.

Sincerely yours

Corresponding author: Chih-Wei Tsao, M.D., Ph.D.

Division of Urology, Department of Surgery,

Tri-Service General Hospital,

No. 325, Section 2, Cheng-Gung Road,

Neihu District, Taipei 11490, Taiwan, China.

Telephone: +886-2-87927170            Fax: +886-2-87927172

e-mail: weisurger@gmail.com

Comments of Reviewer 1 to Author:

It is a well-designed paper with correct methodology. The manuscript is clear and quite complete; however I have some suggestions.

Blood total cholesterol level remained increased compare to control group after metformin treatment. It would be interesting to determine blood levels of very low-density lipoprotein (VLDL), low density lipoprotein (LDL) and high density lipoprotein (HDL) cholesterol. Even oxidation of LDL cholesterol (oxLDL) since it has more affinity for macrophages and induces inflammation. The article would improve with these measurements.

Our response:

We thank the reviewer for the kindly suggestion. Considering of the availability of blood sample from each mice, we have to assign the limited specimen and blood to approachable analysis. We will take your suggestion in our further study.

Increased testicular cholesterol in HFC group was not reverted with metformin treatment. There is a decreasing trend without statistical significance. Do the authors have any explanation? Can it be related to treatment concentration or duration? Have the authors performed experiments with a higher metformin concentration or treatment duration? Maybe, in that case, this trend would be significant. 

We response:

We thank you for the beneficial comment. We have further explanation on Lines 169-172: 「There were no statistical differences of daily food intake among the individual groups (NC: 4.1±0.6 g, HFC: 3.3±0.6 g, HFC+Met: 2.6±0.3 g). To evaluate the metabolic effect of metformin, 0.05% metformin was administered for 8 weeks and the actual intake was about 30 mg/kg/day calculated by actual intake. 」

Kim et al [2016 Eun Kyung Kim et al., Mediators Inflamm (https://pubmed.ncbi.nlm.nih.gov/27057099)] show that 50 mg/kg/day metformin administrated by daily gavage for 14 weeks had markedly reduced cholesterol concentration compared to the group feed with 10 mg/kg/day metformin in C57BL/6 mice.

In our study, Figure 1b shows cholesterol levels increased in group HFC and HFC+ Met compare to control, however, after metformin treatment, the level decreased in the HFC+ Met compared to HFC group. Bars with different letters indicate that values are significantly different. ( p < 0.05) All the figure legends have been revised and the changes were highlighted in red color.

Therefore, the concentration and duration of metformin treatment could have the trend to affect the lipid profiles in high-fat diet-induced obese mice.

Specific comments

Metformin treatment increased antioxidant capacity in the testis, but MDA continued higher compared to control. Do the authors have any explanation? Can it be related to treatment concentration or duration?

Although metformin increased some antioxidant defenses in the testis, it was not able to revert the increment of MDA levels induced by high fat diet as figure 6 shows. Can the authors explain why? How do the authors conclude the metformin has an antioxidant capacity in lines 31 and 370 if treatment with metformin did not decrease malondialdehyde content to control levels (figure 6)?

Lines 357-358: “The results showed that metformin ameliorated the imbalance in the antioxidant system and decreased the testicular lipid peroxidation (MDA) level in the HFC-fed obese mice” Is this of statistical significance? Figure 6 shows MDA levels increased in HFC+ Met compare to control.

Our response:

We thank the reviewer for the academic comment. Figure 6 shows MDA levels increased in group HFC and HFC+ Met compared to control, however, after metformin treatment, MDA level decreased in HFC+ Met compared to HFC group. Bars with different letters indicate that values are significantly different. ( p < 0.05) All the figure legends have been revised and the changes were highlighted in red color.

Spermatozoa spontaneously produce a variety of reactive oxygen species (ROS) and have an enzymatic antioxidant system to protect themselves against oxidative stress. The antioxidant system is a dynamic process, lots of factors involved in such as the different species, various sources of stress, dietary intervention which lead to the different amount or activity of each enzyme. In our study, the Figure 6 shows that SOD activity is the same in control and HFC, upon our knowledge, the excess superoxide (O2- ) can’t be convert to H2O2 and reflected to the elevated MDA level in HFC compared to the control group. The HFC + met group has higher SOD activity than HFC group, combined with elevated CAT and GPx, which decreases the testicular MDA level.

Similar experimental design of Jifeng Ye’s study concluded that HFC+ Met could decreased MDA content compared to HFC, and metformin treatment reduced MDA even under control level [2019 Jifeng Ye et al., Oxidative Medicine and Cellular Longevity (https://pubmed.ncbi.nlm.nih.gov/31583050)]. However the metformin treatment of Jifeng Ye’s study was administered with 200 mg/kg/day calculated by actual intake, which concentration was 6.66 fold compared to our study (30 mg/kg/day).

Abstract: Line 27 “superoxide peroxidase”. It should be written superoxide dismutase.

Our response:

We thank the reviewer to point this out. It has been revised and the changes were highlighted in red color on Line 26.

Introduction: Lines from 63 to 71. It is general knowledge. This paragraph should be shortened. Moreover, refence 22 is not appropriate.

Our response:

We thank the reviewer to point it out. It has been revised and the changes were highlighted in red color on Lines 61 to 66. The reference 22 has been corrected to 21 (https://pubmed.ncbi.nlm.nih.gov/22217824).

Could please the authors write a hypothesis at the end of the introduction?

Our response:

We thank the reviewer to point this out. It has been revised and the changes were highlighted in red color on Lines 78 to 80.

Materials and Methods: Line 100 (HFC+Met, n=15), however in the abstract line 21 (HFC+Met group, n=16) Which one is the right number?

Our response:

We thank the reviewer to point it out. It has been revised and the changes were highlighted in red color on Line 21. The N number is 15, this is our typo and we feel regretful for this mistake.

Results: Lines 241 and 242. At the end of the sentence: “A similar decreasing trend was also observed in the testicular cholesterol level, but this did not reach statistical significance” should indicate “table 2”.

Our response:

We thank the reviewer to point this out. It has been revised and the change was highlighted in red color on Lines 242.

Lines 264 and 265: The results showed that the SOD, CAT and GPx activities in the metformin-treated mice were higher than in the high-fat diet-fed mice. However, SOD activity is the same in control and HFC. Do the authors have any explanation?

Our response:

We thank the reviewer for the professional comment.

Oxidative stress is an important inducer which is associated with male infertility and causes the testicular oxidative stress leads to an increase in germ cell apoptosis and subsequent hypospermatogenesis.

Spermatozoa spontaneously produce a variety of reactive oxygen species (ROS) and have an enzymatic antioxidant system to protect themselves against oxidative stress. The antioxidant system is a dynamic process, lots of factors involved in such as the different species, various sources of stress, dietary intervention which lead to the different amount or activity of each enzyme. In our study, the figure 6 shows that SOD activity is the same in control and HFC, upon our knowledge, the excess superoxide (O2- ) can’t be convert to H2O2 and reflected to the elevated MDA level in HFC compared to the control group. The HFC + met group has higher SOD activity than HFC group, combined with elevated CAT and GPx, which decreases the testicular MDA level.

Discussion: In line 305, authors wrote: “metformin treatment ameliorated hypercholesterolemia, hyperglycaemia and insulin resistance”. However, serum cholesterol levels continue increased after metformin treatment compared to the control group (Figure 1b). Moreover, testicular cholesterol level is also increased in HFC and HFC + Met (table 2). Authors explain there is a decreasing trend without statistical significance (lines 241 and 242) and conclude in lines 335 and 336 that metformin decreases intratesticular cholesterol concentration. This conclusion is not appropriate since there are no statistical differences, they should improve it.

Our response:

We thank the reviewer to point this out. It has been revised and the changes were highlighted in red color.

Lines 308 to 310 : 「metformin treatment ameliorated hyperglycemia and insulin resistance, the minor improvement of hypercholesterolemia were also observed between the group of HFC and HFC+Met.」

Lines 339-340:「the intratesticular cholesterol concentration was increased in the obese group, and decreased after metformin treatment but not reverted to normal.」

Lines 361-362: “The use of metformin may activate the AMPK pathway, increasing alanine production and thus promoting the antioxidant capacity in the testis [55]”. The authors did not analyse this pathway, and it could be interesting since AMPK is a negative regulator of NADPH oxidase. Moreover, PPARg can increase AMPK. The authors measured PPARg protein expression and they did not find changes. The article would benefit from measurement of AMPK or NAPDH oxidase protein level and elucidate the antioxidant capacity of metformin.

Our response:

We thank the reviewer to point it out. Thank you for your kindly suggestion.

According to the study by Qing-Qing Min et al., metformin treatment (200 mg/kg/day) improved HFD-induced obesity and lipid metabolism by the downstream molecules of AMPK. Considering of the availability of blood sample from each mice, we have to assign the limited specimen and blood to approachable analysis. In this study, mice were treat metformin 30 mg/kg/day, the related low dose of treatment and we will take your suggestion in our further study.

Min, Q.-Q.; Qin, L.-Q.; Sun, Z.-Z.; Zuo, W.-T.; Zhao, L.; Xu, J.-Y. Effects of Metformin Combined with Lactoferrin on Lipid Accumulation and Metabolism in Mice Fed with High-Fat Diet. Nutrients 2018, 10, 1628. doi: 10.3390/nu10111628

Reviewer 2 Report

The paper Metformin ameliorates testicular function and spermatogenesis in male mice with high-fat and  high-cholesterol diet-induced obesity  by Chin-Yu Liu et al., describe the results of treating obese mice with metformin in fertility parameters (testicular function, spermatogenesis) and they also investigate some other biomarkers of oxidative stress, inflammation and others. The conclusion of this work is that metformin ameliorates obesity-induced reduced testicular spermatogenesis and semen quality by increasing testosterone levels and reducing oxidative stress. The results would be interesting, if they were more robust. Also, if they were the paper would need some clarifications and changes before consideration for publication.

Major issues:

1-The paper describes a list of negative results: 1) among metabolic markers, only glucose and HOMA-IR show amelioration on mice treated with metformin; 2) Liver markers: no change in any marker; 3) No change sin testicular morphology, weight, etc. and only a tiny rescue of sperm motility and morphology; 4)  so there is no way to understand why there is a rise in testosterone levels, which is believed to play a role in amelioration of fertility.

With inflammatory and pro-apoptotic markers things goes the same way. Very little changes.

2-The results are difficult to follow with the current writing of the statistics. For example, in all figures, in which the authors have shown statistical significance (SS), they use letters in lowercase to stand for SS. I have nothing against this, but I do not known each letter what refers to. Also, they say SS corresponds to p-values lower than 0.05. Do this mean that all the p-values obtained are between 0.01 and 0.05? So please, asterisk and bars, to relate each comparison with the appropriate control, or an in depth explanation in the figure legend. Also, there is no harm in writing the p-values in the text.

More on this issue, N numbers are not very clear. When they show average±SD do this mean for the 10 animals in the control group (for example)? Do they do each average with all animals within a group? Do they do several technical replicas from one animal, or they take just one value from each mice? All this needs to be crystal clear, to be able to understand the changes observed among the groups.

3- Testosterone levels seems to be going up in metformin-treated mice in this work. This agrees with previous data by Annie et al., (https://www.ncbi.nlm.nih.gov/pubmed/32249489), described in  diabetic mice. However, this is in conflict with what has been observed in humans. In this paper, Ozata et al. (https://www.ncbi.nlm.nih.gov/pubmed/11707532), have observed that metformin treatment induces reduction of circulating testosterone in type 2 diabetics. How do the authors reconcile these findings? Are mice good models to investigate these issues, if they do not reproduce what it has been observed in humans?

4-There are many conflicting results with previous literature. For example, the one described above. But also

 Minor points:

-Figure sections appear in lower case in the panels, while in the figure legend they appear in Uppercase.

-Tittles in the results section are weird. Usually, these titles describe a result, not the sample they are analyzing. For example, 3.2. Liver weight, liver weight to body weight ratio, hepatic histological evaluation and hepatic function. Why they don’t write: Metformin treatment do not reverse liver weight nor liver steatosis in obese mice, for example. There is no need to hide negative results. It is what it is. The same apply to many other titles.

- This manuscript may be improved with some English editing. For example, Line 331:..”increasing body weight. Appropriate hormones levels are crucial for male reproductive capacity. The t Testicular testosterone concentration plays an important part in spermatogenesis.

Etc….

- The phrase “Further mechanistic analyses..” in the abstract is a bit optimistic, since I cannot see any sign of mechanistic analysis in the paper.

Author Response

Assistant Editor

Nutrients

Dear Professor Nemanja Marinkovic:

Please find enclosed our revised original paper entitled “Metformin ameliorates testicular function and spermatogenesis in male mice with high-fat and high-cholesterol diet-induced obesity”. We appreciated the comments and suggestions provided to further improve our manuscript.

Sincerely yours

Corresponding author: Chih-Wei Tsao, M.D., Ph.D.

Division of Urology, Department of Surgery,

Tri-Service General Hospital,

No. 325, Section 2, Cheng-Gung Road,

Neihu District, Taipei 11490, Taiwan, China.

Telephone: +886-2-87927170            Fax: +886-2-87927172

e-mail: weisurger@gmail.com

Comments of Reviewer 2 to Author:
The paper Metformin ameliorates testicular function and spermatogenesis in male mice with high-fat and  high-cholesterol diet-induced obesity  by Chin-Yu Liu et al., describe the results of treating obese mice with metformin in fertility parameters (testicular function, spermatogenesis) and they also investigate some other biomarkers of oxidative stress, inflammation and others. The conclusion of this work is that metformin ameliorates obesity-induced reduced testicular spermatogenesis and semen quality by increasing testosterone levels and reducing oxidative stress. The results would be interesting, if they were more robust. Also, if they were the paper would need some clarifications and changes before consideration for publication.

Major issues:

1-The paper describes a list of negative results: 1) among metabolic markers, only glucose and HOMA-IR show amelioration on mice treated with metformin; 2) Liver markers: no change in any marker; 3) No change sin testicular morphology, weight, etc. and only a tiny rescue of sperm motility and morphology; 4)  so there is no way to understand why there is a rise in testosterone levels, which is believed to play a role in amelioration of fertility.

With inflammatory and pro-apoptotic markers things goes the same way. Very little changes.

Our response:

We thank the reviewer for the helpful comment. Our study results showed that 1) statistically significant amelioration in serum cholesterol, glucose and HOMA-IR & trend of decrease in serum triglyceride level after treatment with metformin [Figure 1]; 2) trend of decrease in serum AST and ALT after treatment with metformin [Figure 2]; 3) statistically significant increment in sperm motility, normal sperm morphology and mean testicular biopsy score (MTBS) & trend of enhancement in thickness of germinal epithelium (GE) after treatment with metformin [Figure 3 & 4]; 4) statistically significant elevation in 17 β-HSD of steroidogenesis-related protein and testosterone level after treatment with metformin [Figure 5 & Table 2], and the result is consistent with V.U. Nna’s study [2019 V.U. Nna et al., Andrology (https://pubmed.ncbi.nlm.nih.gov/30515996)]. Although the inflammatory and pro-apoptotic markers of our study were not manipulated after treatment with metformin, the oxidative stress is an important inducer which is associated with male infertility and leads to increase of germ cell apoptosis, then subsequently inducing hypo-spermatogenesis. Spermatozoa spontaneously produce a variety of reactive oxygen species (ROS) and have an enzymatic antioxidant system to protect themselves against oxidative stress. The antioxidant system is a dynamic process, lots of factors involved in such as the different species, various sources of stress, dietary intervention which lead to the different amount or activity of each enzyme. In our study, the Figure 6 shows that SOD activity is the same in control and HFC, upon our knowledge, the excess superoxide (O2- ) can’t be convert to H2O2 and reflected to the elevated MDA level in HFC compared to the control group. The HFC + met group has higher SOD activity than HFC group, combined with elevated CAT and GPx, which decreases the testicular MDA level.

2-The results are difficult to follow with the current writing of the statistics. For example, in all figures, in which the authors have shown statistical significance (SS), they use letters in lowercase to stand for SS. I have nothing against this, but I do not known each letter what refers to. Also, they say SS corresponds to p-values lower than 0.05. Do this mean that all the p-values obtained are between 0.01 and 0.05? So please, asterisk and bars, to relate each comparison with the appropriate control, or an in depth explanation in the figure legend. Also, there is no harm in writing the p-values in the text.

Our response:

We thank the reviewer to point this out. All the figure legends have been revised and the changes were highlighted in red color. Data were used to identify which pairs of treatment means differ by one-way ANOVA and Fisher's protected least significant difference test. Bars with different letters indicate that values are significantly different. ( p < 0.05) Bars with the same letters indicate that values are not significantly different. ( p > 0.05)

Take an example of serum AST of Figure 2e, there were no significant differences between the groups of HFC and HFC+Met (Bars with “a” or “ab”). The groups of HFC+Met and NC (Bars with “b” or “ab”) also showed the same values in statistics. The NC group (Bar with b) was significantly reduced in comparison with HFC or HFC+Met groups.

More on this issue, N numbers are not very clear. When they show average±SD do this mean for the 10 animals in the control group (for example)? Do they do each average with all animals within a group? Do they do several technical replicas from one animal, or they take just one value from each mice? All this needs to be crystal clear, to be able to understand the changes observed among the groups.

Our response:

As described in the Experimental design (line 91-96), all the male C57BL/6 mice were randomly divided into three groups: the NC group (n = 10), the HFC group (n = 15) and HFC+Met group (n = 15). The N number of semen analysis (Figure 3) is 9 (NC group) to 15 within a group (HFC and HFC+Met group). Considering the limitation of a few specimen from each mice, the N numbers of serum analysis (Figure 1) is 4 (NC group) to 8 (HFC and HFC+Met group) within a group; the hepatic histology is 6 (Figure 2) and testicular histology (Figure 4) is 5 within a group; the testis related analysis (Figure 5 to Figure 8) are 6 (NC group) to 9 (HFC and HFC+Met group) with duplicate experiments. Each mice has been checked sperm quality analysis but the testis were assigned to histological exam or other analysis.

3- Testosterone levels seems to be going up in metformin-treated mice in this work. This agrees with previous data by Annie et al., (https://www.ncbi.nlm.nih.gov/pubmed/32249489), described in  diabetic mice. However, this is in conflict with what has been observed in humans. In this paper, Ozata et al. (https://www.ncbi.nlm.nih.gov/pubmed/11707532), have observed that metformin treatment induces reduction of circulating testosterone in type 2 diabetics. How do the authors reconcile these findings? Are mice good models to investigate these issues, if they do not reproduce what it has been observed in humans?

Our response:

We thank the reviewer for these advanced suggestion. Our study results showed that the similar conclusion of metformin treatment increasing testosterone level and spermatogenesis stage as most reference studies focusing the metformin effect in obese rats and mice [2015 Wen-Jie Yan et al., J Assist Reprod Genet (https://pubmed.ncbi.nlm.nih.gov/26081124); 2019 V.U. Nna et al., Andrology (https://pubmed.ncbi.nlm.nih.gov/30515996); 2020 Nicole O McPherson and Michelle Lane, Asian Journal of Andrology (https://pubmed.ncbi.nlm.nih.gov/32098932)]. However the reviewer reported that the study of Ozata et al. has the converse result, metformin treatment leading to reduced free testosterone and decreased total testosterone levels in obese men. We ever explored the content of Ozata’s study and found all the subjects either in diabetic or nondiabetic obese groups were near overweight category. Besides the varied testosterone and leptin levels of Ozata’s study showed significant decrease after metformin treatment in all subjects. No reported values of HOMA or lipid profiles were available in the study. Moreover reported results of Ozata’s study were inconsistent with other human study with metformin treatment [2010 Casulari L.A. et al. Minerva Endocrinologica (https://pubmed.ncbi.nlm.nih.gov/20938417); 2011 Giuseppe Morgante et al. Fertlity and Sterility (https://pubmed.ncbi.nlm.nih.gov/21194687)], both disclosed metformin treatment enhanced testosterone levels and semen quality in metabolic syndrome males. Another human study of E Bosman et al. declared that infertile hyperinsulinaemic men can benefit from metformin treatment and should be advised on the use of nutritional supplements with antioxidant properties [2015 E Bosman et al. Andrologia (https://pubmed.ncbi.nlm.nih.gov/25359661)]. Herein combining the above rat/mice and human studies results supported our study conclusion, metformin treatment has the potential to activate targets resulting in an overall improvement of fertility in obesity or high fat diet induced infertile subjects.

4-There are many conflicting results with previous literature. For example, the one described above. But also Minor points:

-Figure sections appear in lower case in the panels, while in the figure legend they appear in Uppercase.

Our response:

We thank the reviewer to point this out. All the figure legends have been revised and the changes were highlighted in red color.

-Tittles in the results section are weird. Usually, these titles describe a result, not the sample they are analyzing. For example, 3.2. Liver weight, liver weight to body weight ratio, hepatic histological evaluation and hepatic function. Why they don’t write: Metformin treatment do not reverse liver weight nor liver steatosis in obese mice, for example. There is no need to hide negative results. It is what it is. The same apply to many other titles.

Our response:

We thank the reviewer to point it out. All the titles have been revised and the changes were highlighted in red color.

- This manuscript may be improved with some English editing. For example, Line 331:..”increasing body weight. Appropriate hormones levels are crucial for male reproductive capacity. The t Testicular testosterone concentration plays an important part in spermatogenesis.

We response:

We thank the reviewer to point this out. All the figure legends have been revised and the changes were highlighted in red color in Line 336-337.

- The phrase “Further mechanistic analyses..” in the abstract is a bit optimistic, since I cannot see any sign of mechanistic analysis in the paper.

We response:

We thank the reviewer to point it out. We have deleted the statement of the manuscript in Line 26.
